# Effect of Rolling Resistance Model Parameters on 3D DEM Modeling of Coarse Sand Direct Shear Test

**DOI:** 10.3390/ma16052077

**Published:** 2023-03-03

**Authors:** Mohamed Amine Benmebarek, Majid Movahedi Rad

**Affiliations:** Department of Structural and Geotechnical Engineering, Faculty of Architecture, Civil Engineering and Transport Sciences, Széchenyi István University, Egyetem tér 1, 9026 Gyor, Hungary

**Keywords:** DEM, direct shear box, contact model, sensitive analyses, peak shear stress, residual shear stress

## Abstract

This paper deals with the micro and macro behaviors of coarse sand inside a direct shear box during a geotechnical test. A 3D discrete element method (DEM) model of the direct shear of sand was performed using sphere particles to explore the ability of the rolling resistance linear contact model to reproduce this commonly used test considering real-size particles. The focus was on the effect of the interaction of the main contact model parameters and particle size on maximum shear stress, residual shear stress, and sand volume change. The performed model was calibrated and validated with experimental data and followed by sensitive analyses. It is shown that the stress path can be reproduced appropriately. For a high coefficient of friction, the peak shear stress and volume change during the shearing process were mainly affected by increasing the rolling resistance coefficient. However, for a low coefficient of friction, shear stress and volume change were marginally affected by the rolling resistance coefficient. As expected, varying the friction and rolling resistance coefficients was found to have less influence on the residual shear stress.

## 1. Introduction

The direct shear test (DST) is a conventional laboratory test widely used in geotechnical investigations to determine the shear strength and dilatancy of granular materials such as soil, rock, and powder. Despite this test being widely used, understanding the micro-behavior of localization is still lacking. Direct shear has been extensively studied in detail at the macroscopic level, using continuum models via the finite difference method or finite element method with elasto-plastic [1,2] or hypoplastic [3] behavior. However, using the continuum constitutive model, it is challenging to consider granular material properties such as the grain size, grain shape, roughness, and porosity, which are crucial parameters when studying the mechanical behaviors of granular materials.

In the last few years, the discrete element method (DEM) has become the most promising numerical tool for modeling strongly discontinuous, heterogeneous, and nonlinear granular materials [4,5,6,7,8,9,10,11,12]. The DEM, in which granular material is represented as particles assembly interacting with each other, has great advantages for better understanding the mechanical behaviors of geotechnical materials from a meso-mechanical point of view and provides a good insight into the deformation of granular material properties and shear strength.

Therefore, several studies have been conducted to develop numerical DEM methods considering 2D [13,14,15,16,17,18,19,20] and 3D [4,8,9,10,21,22,23,24] particles under different loading conditions, such as the direct shear test, which is a widely used conventional laboratory test in geotechnical investigations [25,26,27,28,29,30,31,32,33,34]. 

Cui and O’Sullivan [4] were the first to compare physical test data using perfect metal spheres with direct shear test DEM models. Zhou et al. [6] examined the scaling behavior in direct shear tests. They focused on the dependence of the shear band on the particle size, the dependence of the bulk friction on the particle size, and the formation of the shear band. In addition, they confirmed that the box height and length influence the bulk friction, as observed in a study using physical tests [35]. However, their analysis is limited to 2D calculations, and only horizontal displacements were examined. Yan and Ji [36] compared their results from DEM with direct shear tests on irregular limestone rubbles. In their method, to simulate the shape of the real rubbles, they used clumps. Both studies obtained a good agreement between real test data and DEM results. Härtl and Ooi [37] used the Jenike direct shear test to explore the microscopic friction on the bulk friction and the effect of the particle shape. They used single spherical particles and a clump of two overlapping spheres to investigate the effect of the shape. According to their results, the packing density has less effect on the bulk friction than the interlocking of the particles. Kim and al. [38] studied the effect of the opening between shear boxes on the shear behavior in the direct shear box test. Their results showed that the opening size influences the dilatancy for all soil samples and the peak shear stress. Salazar et al. [8] performed a 3D DEM simulation of the direct shear test of sand using a rolling friction model to include the grain shape of the sand with sizes corresponding to the actual grain size distribution. They showed that the dilatancy was challenging to reproduce while the stress path could be accurately replicated. In addition, they concluded that by comparing the accuracy of the results with the experimental data, the unscaled material appears to be the best model for replicating the shear test. However, a parallel grading curve could be considered when the computational cost is essential. Nitka and Grabowski [39] performed a 3D DEM simulation using spherical grains and carefully studied the behavior of sand grains before the peak. The localization characteristics and grain-scale phenomena were captured using the actual size of the particles. They concluded that it is possible to observe and predict the phenomena inside the localization zone early. Despite the fact that much research has been done on the sensitivity of shear stress to the parameters, the formulation and localization evolution are not yet well recognized.

In this paper, a novel 3D DEM model of the direct shear test was developed using sphere particles to explore the ability of the rolling resistance linear contact model to reproduce this widely used test, taking into consideration the actual size of particles. After calibrating the developed model with the experimental laboratory data provided by Salazar et al. [8], the effect of the model’s micro-mechanical parameters and particle size on the peak shear stress and the residual stress as well as the sand dilatancy during shearing was studied with a sensitivity analysis.

This paper’s novelty lies in which micro-mechanical parameters are responsible for controlling the peak and residual shear stress behaviors., as well as the interaction between friction and rolling resistance coefficients on the shear stress path. The simulation is run several times with different friction and rolling resistance coefficient values to investigate their effect on the shear stress path and the volume change. The flow chart of this research is shown in Figure 1.

## 2. Model Calibration

To develop a numerical model and to determine material micro-parameters, 3D DEM simulations were performed to model the experimental results of direct shear tests.

### 2.1. Laboratory Direct Shear Tests

In the direct shear test, a soil sample is placed in a box. The box is divided into two halves: upper and lower. The upper section is fixed, whereas the lower section is pushed or pulled horizontally relative to the lower section. A confinement force is applied to the top wall, which is free to move vertically. As the sideways force is increased, the sample will eventually fail or ‘shear’ along the horizontal plane, and the force required results in the material’s shear strength (as shown in Figure 2).

This article used data from direct shear tests conducted by Salazar et al. [8] to calibrate and validate the numerical DEM model by obtaining the relationship between the micro and macroscopic parameters used in the model. 

Salazar et al. [8] have performed a series of direct shear tests on a specimen of $60\times 60\times 24{\;\mathrm{mm}}^{3}$ of an angular non-cohesive coarse sand subjected to different confinement pressures of 160 kPa, 80 kPa, and 40 kPa. In order to acquire dilation curves and shear stress, the laboratory test was instrumented using force and displacement transducers. The sand has 1% of fines (a diameter less than 0:075 mm) and the majority of the mass is retained in sieve N 16, which contains larger grains (between 2.16 mm and 1.18 mm in diameter). Its main properties are listed in Table 1.

### 2.2. 3D DEM Simulation of DST

The DEM model of the DST in this study was made using the three-dimensional particle flow code (PFC3D) from Itasca (Minneapolis, MN, USA) and was based on soft contact and rigid body approaches. A virtual reproduction of the actual direct shear test was performed to reproduce the direct shear test with a DEM model. In this study, the numerical model’s dimensions, boundary conditions, and materials are based on direct shear tests made by Salazar et al. [8].

The DEM model setup of this direct shear box with dimensions 60 × 60 × 24 mm^3^ is presented in Figure 3. Ten rigid wall elements enclose the numerical model of the direct shear apparatus and are horizontally split into two equal halves (upper and lower shear boxes). The lower shear box is open at the top, and the upper shear box is open at the top and bottom sides.

The top wall of the upper shear box can move up and down during compression and shearing. During the shearing process, the upper box remains fixed during the whole test, while the lower box moves horizontally. Two more walls (one on the left and one on the right) are used to close the horizontal surfaces on either side of the shear plane to prevent particles from dropping out of the shear box. This is an essential step to prevent particles from escaping the box during shearing. The recorded reaction force depends on the properties of the shear band created by the relative displacement between the two rigid half-boxes.

The particle shape significantly affects the mechanical behavior of a granular solid and costs a lot of storage capacity and computing time [40,41,42]. Nevertheless, several researchers have shown that by using sphere particles with a certain rolling friction, it is also possible to achieve the same effect as the non-spherical particles by drastically reducing the storage capacity and computing time [22,32,43].

Therefore, to allow for capturing the localization of a huge number of particles and to reduce the calculation costs, the granular material is simulated in the form of sphere assemblies with diameters depending on the grain size distribution of the sand tested.

The particle size is controlled by predefined minimum and maximum diameters (Dmin = 1.2 mm and Dmax = 2.15 mm), ensuring uniform particle-size distribution.

There are three steps in the numerical simulation of the direct shear test: specimen making, consolidation, and shearing.

To generate an equilibrium specimen, the box must first be defined. Then, balls were generated inside the box. For the generation of balls, the PFC porosity command was used, which allows for matching the target porosity by randomly generating balls in the specified box regardless of an overlap. The rolling resistance linear contact model included in PFC3D was selected to define the local contact ball–ball and ball–wall to account for the grain roughness (the particle shape effect). Several calculations were needed until an equilibrium state was reached. Since we are interested in the final configuration, the local damping coefficient was defined during this simulation to remove the kinetic energy from the system effectively.

After equilibrium, a predefined normal stress was applied to the shear box’s top wall until the sample was consolidated via a servo controller loading mechanism, which controls the velocity of the top wall until the reaction force in the vertical direction on the upper wall reaches the target stress values as used in laboratory tests (160 kPa, 80 kPa, and 40 kPa).

Finally, the shearing process started once the consolidation was completed. A displacement was imposed on the lower half at a constant loading rate of 0.01 m/s to achieve a target horizontal displacement of 8 mm.

The loading rate was considered slow enough to guarantee that the test was performed under quasi-static conditions. 

The response of a DEM model is sensitive to the loading rate, which must be sufficiently slow to ensure a quasi-static response. Therefore, a loading velocity sensitivity analysis was performed for the lower box velocity by adjusting the peak shear loading force and the ratio of the unbalanced force magnitude to the applied force magnitude. Considering the quasi-static condition and computational efficiency, the selected loading velocity applied to the lower box was set at 0.01 m/s, resulting in an unbalanced force to contact force ratio of less than 0.001 during the shearing process. By decreasing the velocity further, the peak and residual shear stress results were not affected, but the computation time became troublesome.

For this step, the simulation was performed multiple times with different friction and rolling resistance coefficient values to study their effects on the shear strength and vertical displacement. 

The normal and shear stresses $\sigma_{n}$ and $\sigma_{s}$ acting on the shear plane are calculated using the equations below:(1)$$\sigma_{n}=\frac{F_{N}}{D\left(B-\vartheta t\right)}$$
(2)$$\sigma_{S}=\frac{F_{S}}{D\left(B-\vartheta t\right)}$$
where: 

$F_{N}$ is the normal force acting on the shear plane. It is equal to the normal load applied to the sample, $F_{s}$ is the shear force equal to the horizontal forces in the upper box, D and B are the length and width of the shear box, ϑ is the constant rate of the lower box displacement, and t is the duration of the test.

### 2.3. Contact Model

In the DST model, the rolling resistance contact model (with simple particles) was used. The rolling resistance is applied by adding rolling friction at contacts between modelled sand particles, as shown in Figure 4. Compared with the widely used linear contact model, it is better at providing a realistic performance of coarse sand assemblies by restricting a relative particle rotation.

The rolling resistance contact model in PFC3D is based on the linear model but includes a rolling resistance mechanism. The force–displacement law for the rolling resistance linear model updates the contact force and moment as follows:(3)$$F_{c}=F^{l}+F^{d}\;,\;M_{c}=M^{r}$$
where $F^{l}$ denotes the linear force, $F^{d}$ denotes the dashpot force, and $M^{r}$ denotes the rolling resistance moment. The linear and dashpot forces are updated the same as in the linear model, while the rolling resistance moment is updated using the steps below. First, the rolling resistance moment is incremented as:(4)$$M^{r\;}∶=M^{r}-k_{r}∆\theta_{b}$$
where $∆\theta_{b}$ is the relative bend–rotation increment and $k_{r}$ is the rolling resistance stiffness, defined as:(5)$$k_{r}=k_{s}{\bar{R}}^{2}$$
with $\bar{R\;}$, the contact effective radius was defined as:(6)$$\frac{1}{\bar{R\;}}=\frac{1}{R^{\left(1\right)}}+\frac{1}{R^{\left(2\right)}}$$

$R^{\left(1\right)}$ and $R^{\left(2\right)}$ are the radii of the contact particles. If one side of the contact is a wall, the corresponding radius $R^{\left(2\right)}=\infty$.

The magnitude of the updated rolling resistance moment is then checked again compared to the threshold limit:(7)$$M^{r}=\{\begin{array}{c} M^{r},\;∥M^{r}∥\leq M^{*}\\ M^{*}(M^{r}/∥M^{r}∥),\;\mathrm{otherwise} \end{array}$$

The limiting torque is defined as:(8)$$M^{*}=\mu_{r}\bar{R\;}F_{n}^{l}$$
where $\mu_{r}$ is the rolling resistance coefficient and $F_{n}^{l}$ is the normal linear force.

### 2.4. Calibration Results

In PFC, the macroscopic response of the material is derived from the interaction of the microscopic properties input. The objective of calibrating the microscale parameters is to obtain the parameters of the contact particle model in the PFC calculation. 

For DEM rolling resistance linear contact model, four main parameters must be determined: $E^{*}$ (elasticity modulus of grain contact), $K_{n}/K_{s}$ (ratio of normal/shear stiffness of grain contact), μ (friction coefficient of inter-particle), and $\mu_{r}$ (coefficient of rolling resistance). The trial-and-error method was used to solve the problem, in which the main input microscopic parameters were continuously adjusted until the desired macroscopic behavior was reproduced. First, the DEM model was calibrated with Salazar’s results for a normal stress of 160 kPa. The calibrated parameters of the DEM model are summarized in Table 2.

The model better matches the experimental shear stress path for the whole simulation, as shown in Figure 5a. From calibration tests, the initial slope at a low stain stiffness can be controlled by the stiffness of the contact particles and the initial porosity. However, the shear strength can be controlled by particle friction and rolling resistance coefficients. 

After identifying the parameters of the DEM model by calibration, the model was also checked for both confinement pressures of 40 kPa and 80 kPa and compared to the experimental results. The model underestimates the peak value for the 40 kPa and 80 kPa confinements, as observed by Salazar et al. [8]. This might be related to the difficulty of obtaining the same porosity in laboratory shear tests. However, the residual shear stress is precisely reproduced for all the confinements. 

The simulation results of the vertical displacement (Figure 5b), representing the overall volume change (contractive dilatancy), were observed and compared to the experimental data. The shape of the curves is very similar to the response of dense sand, marked by the contractive response for low stain followed by dilatant behavior. As the normal stress increases, the DEM model exhibits the same experimental responses characterized by the contractive response increase and the dilatancy decrease.

Since the rolling friction model is used to improve the shear strength and to approximate the effect of particle shape, it is not possible to sufficiently reproduce the dilatant behavior of the reference soil for a low confinement.

The behavior of the sand in the numerical calculations was close to reality. It is seen that the predicted DEM responses generally agree with the results of the laboratory tests. In general, it can be concluded that the developed model performed reasonably well in capturing the shear behavior of DST under the investigated range of normal stresses. 

## 3. Microscopic Observations

### 3.1. Inter-Particle Forces

The contact forces between the particles are represented as lines whose thickness varies with the force magnitude. The corresponding normal contact force chains that are captured from the front view, illustrating how the applied load was transferred throughout the particle assembly, are shown in Figure 6. This Figure shows the distribution of contact force chains at different shear levels for a DEM simulation test with a normal stress of 160 kPa.

From the relationship between the normal contact force and shear force, it is possible to obtain a better picture of the particles’ overlapping and motion of the particles. From the force chains for the sand sample sheared to a displacement of 0 to 8 mm, it is evident that the contact forces gradually redistribute with an advancing shear. In the initial state, when there is no shear, but normal stress is applied, the contact forces are distributed uniformly within the particle assembly and transferred vertically from the top to the bottom of the shear box. Once the lower box advances to the right, the shearing proceeds until it is close to the peak value. The contact forces intensify from the bottom left to the top right corner, creating a force concentration band that evolves diagonally and becomes more noticeable, meaning that larger normal contact forces are directed diagonally.

At a displacement of 8 mm, the contact force magnitude is the lowest compared to a displacement of 2 mm to 6 mm (during shearing). This can be explained due to a decrease in the coordination number or the average number of contacts of particle linked with an increase in the dilation of particles assembly, and the corresponding decrease in the shear strength (strain softening).

### 3.2. Particles Displacement Vector

Figure 7 shows the evolution of the displacement vectors of the particles within the granular specimen. Particle displacement vectors are provided as illustrated by two independent properties: the force direction and magnitude. Each vector represents the particle’s displacement, with the beginning and end of the vector corresponding to the particle’s initial and final positions and the length of the distance traveled.

The vectors are sketched for 1 of the samples sheared under $\sigma_{n}=160\;\mathrm{kPa}$ to a displacement of 8 mm. The sample shows the displacement of the particles: in the lower box, there are large movements of particles to the right, and in the upper box, there are only slight convex thrusts. Due to the simulations’ choice to move the bottom box to the right, there is a variation in the magnitude of displacement between the lower and upper boxes.

For a displacement from 0–2 mm, in the lower box, particles moved in the horizontal direction, while in the upper box, particles moved downward, resulting in a densification of the particle assembly characterized by a contractive response. There was a large particle displacement arrow in the downward direction at the right edge of the box, meaning that one of the particles moved down due to the void inside the sample and gravity forces.

On the contrary, at a much higher displacement (>2 mm), the shear dilation causes convex thrusts in the upper box. Dilatation occurs because particles in the upper box tend to move upwards. These micromechanical observations clearly show that the continuum mechanics method cannot provide the same level of clarity in the corresponding strain-softening response and the insightful evolution of volumetric changes during shearing within the granular media.

## 4. Sensitivity Analysis

Different samples were modeled and tested to investigate how the main micromechanical parameters affected the macroscopic response. 

### 4.1. Rolling Resistance

The simulation was run multiple times with different rolling resistance coefficient values for two cases of a low and a high friction coefficient to investigate the effect of the rolling resistance on the macroscopic response.

Figure 8 compares shear stress as the increase in the rolling friction coefficient, for µ=0.2 and 0.6. For a low friction coefficient µ=0.2 (Figure 8a), the variation in the rolling resistance coefficient from 0.2 to 0.9 seems to influence shear stress at a large displacement. At a low displacement, the shear stresses are very close. In fact, the rolling friction seems to increase the residual shear stress. Moreover, no peak shear stress is observed, and the shear stress curves increase monotonically until they reach a plateau for horizontal displacements larger than 3–4 mm. For ${µ}_{r}$ values larger than µ, the residual shear stresses are very close and the effect of an increasing µ_r_ is very marginal. From the results of a high friction coefficient µ =0.6 (Figure 8b), increasing the rolling friction coefficient from 0.1 to 0.9 affects small displacements and tends to increase the peak value. In contrast, the large-strain stress value decreases nearly to the same value, probably corresponding to a looser configuration of sand for horizontal displacements larger than 4–5 mm. Residual shear stresses are less influenced by ${µ}_{r}$ compared to peak shear stresses.

From the results of these sensitivity analyses, it can be concluded that it is necessary to choose dense sand with a high µ to represent the typical response of relatively dense sand characterized by peak shear stress and residual shear stress close to the shear stress of loose sand. The increase in the shear peak can be reached by increasing ${µ}_{r}\;$ and µ.

Regarding volume evolution, at a low friction coefficient of µ=0.2, the increase in ${µ}_{r}$ has a small effect on the dilatation curves, which after the contractive response, increase linearly with shearing without considerably affecting the path’s shape. Contradictory to a high friction coefficient µ = 0.6, where dilation increases monotonically with an increasing displacement until reaching a plateau. The results show that increasing the rolling resistance coefficient increases the specimen dilatation.

### 4.2. Friction Coefficient

Furthermore, the effect of the friction coefficient was investigated on a low and a high rolling resistance coefficient. Figure 9 compares shear stress and dilatation curves as a function of the friction coefficient for ${µ}_{r}=0.2$ (Figure 9a) and 0.6 (Figure 9b). Based on the results, increasing the friction coefficient will increase the peak value of shear stress for both a low and a high rolling resistance coefficient. A very low friction compared to the rolling resistance will decrease the residual value. When the friction coefficient is near or higher than the rolling resistance, the residual value reaches a plateau, and the results become closer. The results indicate that it is impossible to adjust the model’s response to large displacements by varying the friction coefficient. By comparing the curve shape of peak shear stresses, the shear peak is reached more quickly by increasing the rolling resistance coefficient. Regarding the volume evolution, increasing the friction coefficient gradually increases the sample dilatation and seems to reach a plateau for a large displacement for high friction and rolling resistance coefficients.

### 4.3. Porosity

There is different shear strength and displacement behavior depending on the soil porosity. A dense sand sample tends to achieve peak shear stress that gradually falls (Figure 10a), reaching a fixed value (residual strength). As the porosity rises, the internal friction angle reduces (there was no peak value detected for the loose samples), but the residual value remains constant. When the initial sample density was dense or medium dense, dilatancy appeared from the start of the test, while for the loose sample, only the contractive response occurred (Figure 10b). As the porosity increases, the dilatancy of the sample gradually decreases. The behavior of the sand in the numerical results was close to reality.

### 4.4. Up-Scaled Effect 

The parallel gradation method was applied to decrease the particle number in the analyses to minimize the computation time and explore the degradation accuracy. Two samples with two and three times the size of the particles were compared to the original studied case. 

According to the results in Figure 11a, the shear stress curves show more fluctuations when larger particles are considered compared to the normal scaled results. It can be explained by the fact that using scaled particles significantly reduces the number of contacts. Any change has a proportionally greater effect on the macroscopic response compared to the unscaled case (original case), where a significant number of contacts exist. For the dilatation (Figure 11b), the particles scaled two times have a similar path to the unscaled ones. Nevertheless, in the case of scaling the sample three times, the dilatation increases sharply.

## 5. Conclusions

In this study, the behavior of coarse sand direct shear test was simulated using 3D DEM. The rolling friction model was used to consider the grain shape’s effect. Different specimens were modeled and tested to further investigate the effect of the micro-mechanical parameters on the macroscopic response. It was observed that the DEM, considering real-size particles, can realistically reproduce experimental macroscopic data. From this investigation, the following conclusions were derived:

In direct shear, the shear stress paths (peak and residual) can be well represented and explored in-depth through the contact force chain and displacement.For a high coefficient of friction (>0.5), the peak shear stress and volume change are mainly affected by increasing the rolling resistance coefficient. However, for a low coefficient of friction (<0.5), the rolling resistance coefficient has a marginal effect on the shear stress and volume change during the shearing process.Varying the friction and rolling resistance coefficients seems to have a negligible influence on the residual shear stress (the contact model parameters interactions lightly affect the residual shear stress).To represent the typical response of relatively dense sand characterized by peak shear stress and residual shear stress close to the shear stress of loose sand, choosing dense sand with a high coefficient of friction is necessary.Using a high rolling resistance, the effect of increasing the friction coefficient is more pronounced for the peak stress and volume change values.The use of the scaling method is the best alternative to reach a reasonable computing time, but it raises the fluctuation of results. This methodology allows for making bigger and more complex models.

## Figures and Tables

**Figure 1 materials-16-02077-f001:**
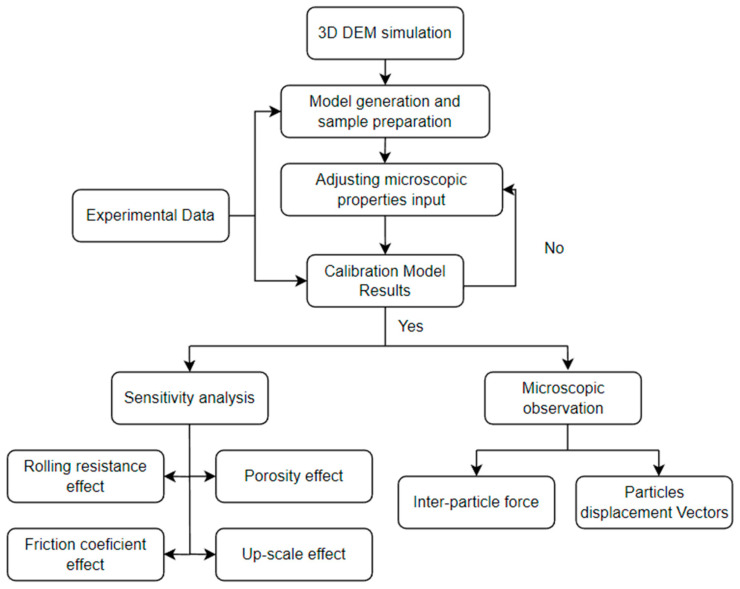
Flow chart of research process.

**Figure 2 materials-16-02077-f002:**
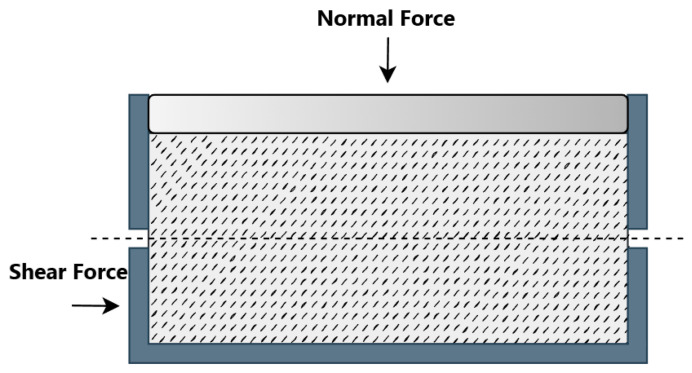
Schematic view of direct shear test.

**Figure 3 materials-16-02077-f003:**
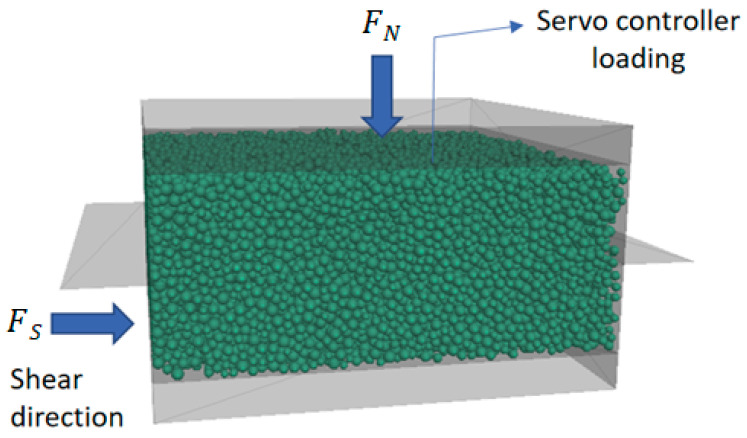
The developed DEM model of the direct shear test.

**Figure 4 materials-16-02077-f004:**
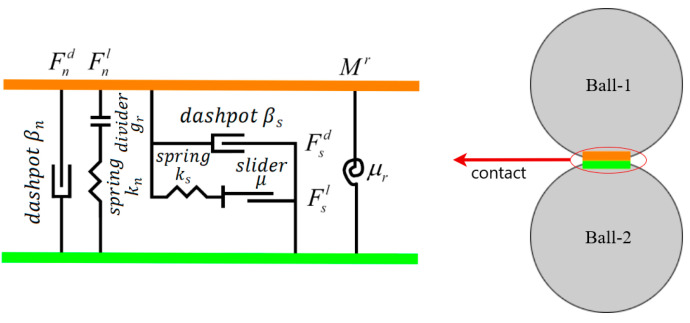
Illustration of the rolling resistance linear model.

**Figure 5 materials-16-02077-f005:**
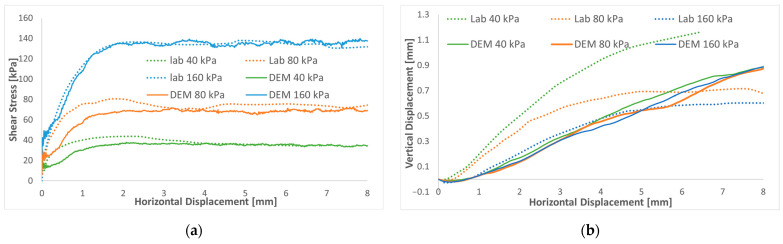
Comparisons between the DEM simulation and experiment (Salazar et al. [8]) for three normal stresses of 160, 80, and 40 kPa: (**a**) shear stress versus shear strain; (**b**) volumetric strain versus shear strain.

**Figure 6 materials-16-02077-f006:**
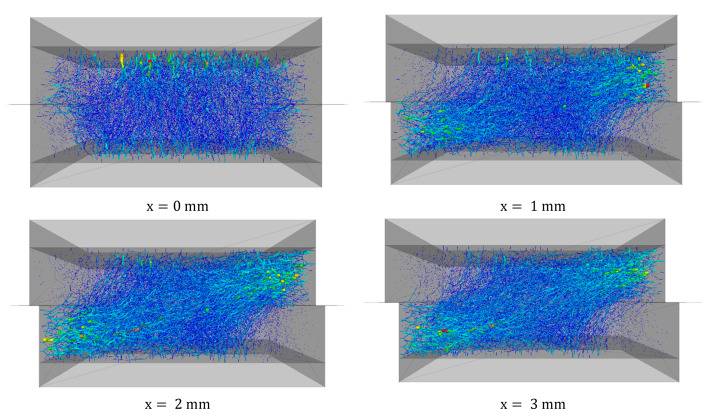
Distribution of contact forces (projection of all contact forces) drawn at the same scale for sample sheared under $\sigma_{n}$ =160 kPa to different distances (x=0−8mm).

**Figure 7 materials-16-02077-f007:**
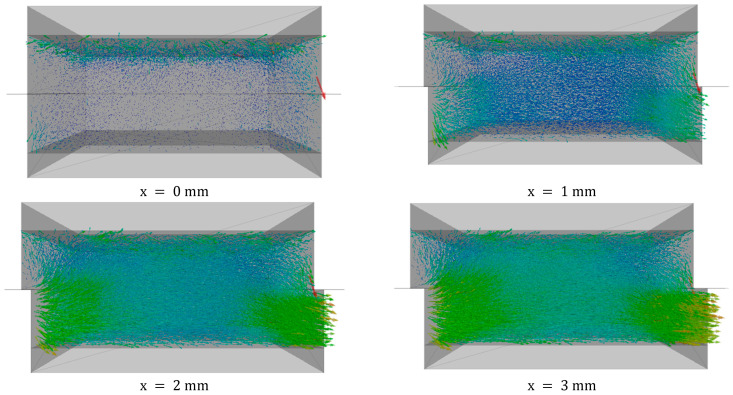
Displacement vectors of particles drawn at the same scale for sample sheared under $\sigma_{n}=160\;\mathrm{kPa}$ to a displacement of x = 8 mm.

**Figure 8 materials-16-02077-f008:**
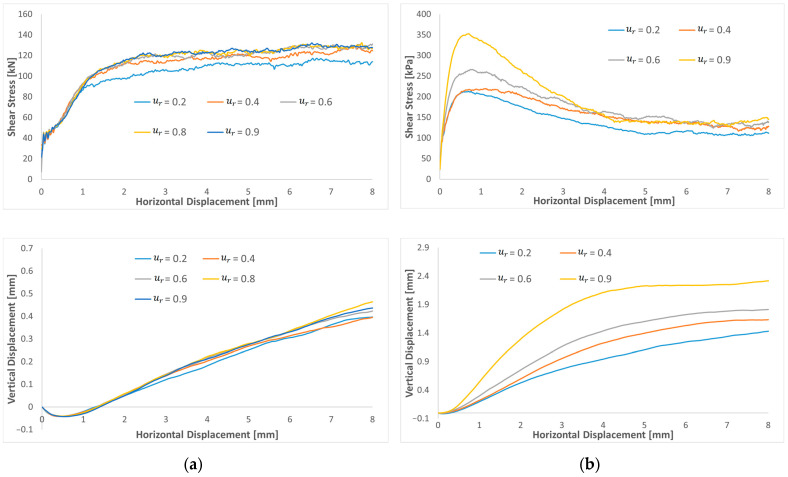
Shear stress and dilatation curves of varying rolling resistance coefficient for a low and a high friction: (**a**) µ=0.2 and (**b**) µ=0.6.

**Figure 9 materials-16-02077-f009:**
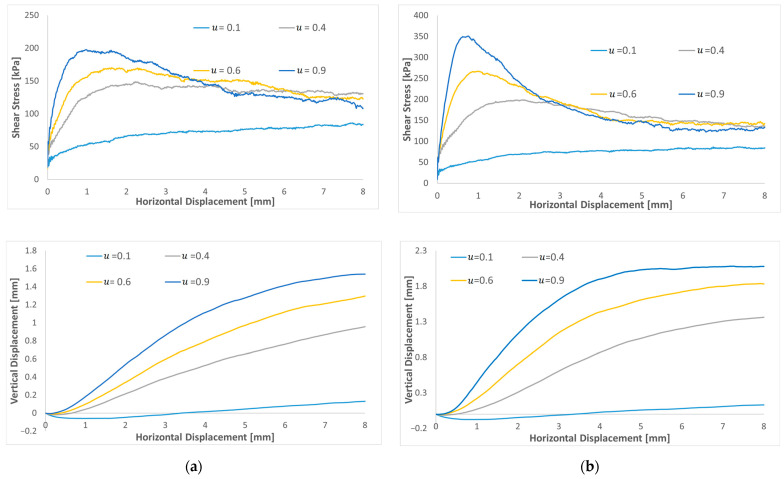
Shear stress and dilatation curves of varying friction coefficient for both high and low rolling resistance friction: (**a**) $\mu_{r}=0.2$ and (**b**) $\mu_{r}=0.6$.

**Figure 10 materials-16-02077-f010:**
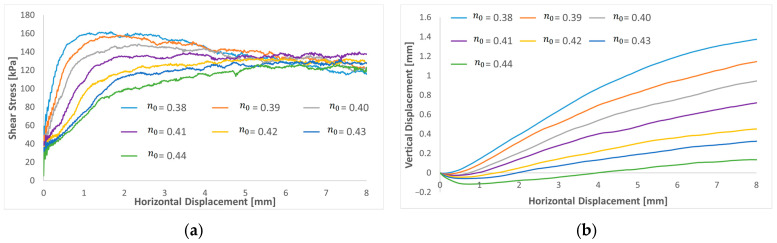
Shear stress and dilatation curves of varying porosity: (**a**) shear stress versus shear strain; (**b**) volumetric strain versus shear strain.

**Figure 11 materials-16-02077-f011:**
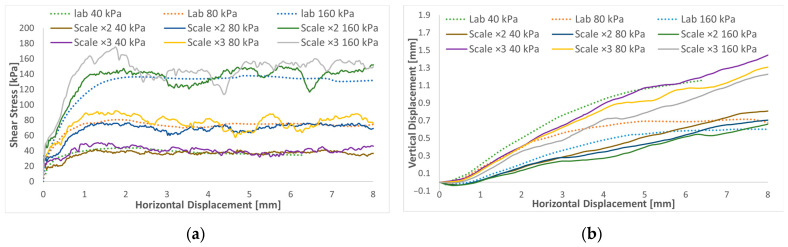
Shear stress and dilatation curves of up-scaled DST tests: (**a**) shear stress versus shear strain; (**b**) volumetric strain versus shear strain.

**Table 1 materials-16-02077-t001:** Sand properties (Salazar et al. [8]).

Property	Value
$\mathrm{Density}:\;\gamma \;\left[\mathrm{kg}/m^{3}\right]$	1578
$\mathrm{Void}\;\mathrm{ratio}:\;e_{0}$	0.679
$\mathrm{Porosity}:\;n_{0}$	0.405
Angle of internal friction: $\varphi^{\prime }$ [°]	35
$\mathrm{Cohesion}:\;c^{\prime }\;\left[\mathrm{kPa}\right]$	8

**Table 2 materials-16-02077-t002:** Microscopic parameters for DEM simulation.

Particles Properties		
Elementary particles size, D	mm	1.2–2.15
$\mathrm{Effective}\;\mathrm{modulus}\;\mathrm{of}\;\mathrm{sand}\;\mathrm{particle},\;E^{*}$	GPa	2
$\mathrm{Normal}-\mathrm{to}-\mathrm{shear}\;\mathrm{stiffness}\;\mathrm{ratio}\;\mathrm{of}\;\mathrm{sand}\;\mathrm{particle}\;K_{n}/K_{s}$	-	1
Density	kg/m^3^	2600
Damping coefficient	-	0.5
Friction coefficient between sand particles, μ	-	0.3
Porosity	-	0.41
Friction coefficient between sand particle and wall, μ	-	0.0
$\mathrm{Rolling}\;\mathrm{resistance}\;\mathrm{coefficient}\;\mathrm{of}\;\mathrm{sand}\;\mathrm{particle},\;\mu_{r}$	-	0.25

## Data Availability

The datasets which are generated during and analyzed in the current study are available in the main manuscript; any additional details can be obtained from the authors.

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
