# Peer review of "Effect of Rolling Resistance Model Parameters on 3D DEM Modeling of Coarse Sand Direct Shear Test"

_materials, 2023, doi:10.3390/ma16052077_

Round 1
Reviewer 1 Report
This paper investigates the micro and macro behaviors of coarse sand under the direct shear test using 3D discrete element method. It is an interesting topic, but Major Revision should be considered before its publication.
1. On the whole, the manuscript is more like a thesis or are port rather than a scientific research. In my understanding, this is not acceptable in a scientific paper of this field of knowledge.
2. My first and primary concern lies in the novelty of this work, as l feel that the novelty issue has not been sufficiently highlighted in the current version. An important question shall be answered: does this work fill up some knowledge gaps which previous articles cannot address?
3. The format of the abstract is not correct. Please read the author guidelines. It is too descriptive. Please add some quantities to it and shorten it through focusing on the main point. In addition, “This research is useful for gaining a better understanding of the mechanical behaviors of the direct shear test.” is not a conclusion.
4. The title of this paper is unavailable, because the rolling resistance model cannot represent all shear behaviors of coarse sand.
5. The direct shear test is a common testing method. Please delete the first paragraph in Introduction section.
6. The discrete element method was widely applied to analyze the mechanical behavior of granular materials, such as
https://doi.org/10.3390/app122010571
https://doi.org/10.1080/10298436.2023.2165650
https://doi.org/10.3390/min13010040
According to the above references, the irregular shape can be simulated in DEM. Why did the author only use the spherical particle and rolling resistance model to simulate the coarse sand?
7. Please provide a flow chart of research process.
8. Please introduces the constitutive model of rolling resistance linear contact model in details.
9. The conclusion part should be more refined to make the findings and contributions of the paper clearer. Furthermore, please note the difference between the conclusions and abstract.
Reviewer 2 Report
This article is generally well presented. However, there are several typos that must be corrected as well as the English language should be improved in various places. A professional English proof reading is recommended.
Some suggestions:
1. Short description and illustration of the DST might be helpful for the reader.
2. Figure 1: define V and P. These notations are not consistent with the ones in Equation 1 and 2.
3. Figure 3 and 4: Legend for the color gradient may be added (values will give the idea of the magnitudes).
4. Was the presence of moisture considered? How can moisture be represented with this approach? A short discussion may be added.
Reviewer 3 Report
The main focus of this paper topic is to develop a 3D numerical model using the Discrete Element Method to simulate the shear behavior of coarse sand under different loading conditions. The model is developed using commercial code and is calibrated using experimental data provided by Salazar et al. [9]. The paper is well organized. In order to further enhance the quality of the paper, the following remarks are given:
1) In the study conducted by Salazar et al. [9], the majority of the grains in the laboratory test are within the [1.18, 2.16] mm size range. Why the authors of the paper use in the numerical model particle size between 1.0 and 1.6 mm? How did they ensure a uniform particle-size distribution in their model?
2) Line 113: Subtitle 2.2.3. should be named: “3D DEM …”
3) Line 163: What is “slow enough” to obtain quasi-static conditions in their simulation? What is criterion for this?
4) In Fig. 2b, there appears to be a significant difference between experimental data and computed vertical displacement. Notably, the differences between the normal loads of 40, 80 and 160 kPa in the numerical model seem to have little impact, as all three computed curves are very close. Based on the findings in Fig. 6, it is possible that adjusting the friction coefficient could yield a more accurate computed vertical displacement that aligns more closely with the experimental data.
5) In Figs. 8a and 8b are more curves than in the legend.
Round 2
Reviewer 1 Report
The submission has been greatly improved and is worthy of publication.
Reviewer 3 Report
Comment to Answer on Remark #1:
The authors just change the grain Dmin and Dmax value in the text - line 162. Did they use in numerical computation previous values of balls size or the new one? These values should have influence on mechanical response of the sand in the numerical model.
